

# Living shorelines achieve functional equivalence to natural fringe marshes across multiple ecological metrics

Robert E. Isdell[1], Donna Marie Bilkovic[1], Amanda G. Guthrie[1], Molly M. Mitchell[1], Randolph M. Chambers[2,3], Matthias Leu[2] and Carl Hershner[1]

[1] Virginia Institute of Marine Science, William & Mary, Gloucester Point, VA, United States of America
[2] Biology Department, William & Mary, Williamsburg, VA, United States of America
[3] Keck Environmental Laboratory, William & Mary, Williamsburg, VA, United States of America

## ABSTRACT

Nature-based shoreline protection provides a welcome class of adaptations to promote ecological resilience in the face of climate change. Along coastlines, living shorelines are among the preferred adaptation strategies to both reduce erosion and provide ecological functions. As an alternative to shoreline armoring, living shorelines are viewed favorably among coastal managers and some private property owners, but they have yet to undergo a thorough examination of how their levels of ecosystem functions compare to their closest natural counterpart: fringing marshes. Here, we provide a synthesis of results from a multi-year, large-spatial-scale study in which we compared numerous ecological metrics (including habitat provision for fish, invertebrates, diamondback terrapin, and birds, nutrient and carbon storage, and plant productivity) measured in thirteen pairs of living shorelines and natural fringing marshes throughout coastal Virginia, USA. Living shorelines were composed of marshes created by bank grading, placement of sand fill for proper elevations, and planting of *S. alterniflora* and *S. patens*, as well as placement of a stone sill seaward and parallel to the marsh to serve as a wave break. Overall, we found that living shorelines were functionally equivalent to natural marshes in nearly all measured aspects, except for a lag in soil composition due to construction of living shoreline marshes with clean, low-organic sands. These data support the prioritization of living shorelines as a coastal adaptation strategy.

## INTRODUCTION

Natural marshes around the world are under assault on myriad fronts. From concerted and ongoing anthropogenic efforts to convert wetlands to "productive" land (e.g., agriculture and aquaculture; *Verhoeven & Setter, 2010*) to accelerating sea level rise (SLR; *Boon et al., 2018*) outpacing sediment accretion (*Kirwan et al., 2010*), salt marshes are changing and disappearing (*Craft et al., 2008*; *Mitchell et al., 2017*). These direct and indirect impacts are not evenly spread across the globe, resulting in some coastal areas experiencing and/or

Corresponding author
Robert E. Isdell, risdell@vims.edu

expecting much greater losses of wetlands than others (*FitzGerald et al., 2008*). From a physical standpoint, marshes in microtidal (≤1-m tide range) settings with both a limited sediment supply and limited opportunities for inland migration are expected to experience the greatest proportional losses (*Kirwan et al., 2010*; *Mitchell et al., 2017*). Losses are likely to be exacerbated where these conditions overlap with extensive watershed development (*Mitchell, Herman & Hershner, 2020*).

Concurrent with the loss of coastal wetlands, SLR increases inundation and erosion of personal property along coastlines. Numerous engineered structures are designed to stabilize a shoreline and prevent erosion and property loss. In the past, shoreline armoring (riprap revetment (riprap, hereafter) and bulkhead) was the primary means to stabilize a shoreline. Whereas both riprap and bulkheads are effective at reducing tidally-driven shore erosion, hardened shorelines are unable to naturally adapt to rising seas, are less resilient during storms, and scour the nearshore sediment through wave refraction (*Gittman et al., 2014*; *Smith et al., 2017*). Ecological studies have consistently found that shoreline armoring negatively impacts the intertidal and nearshore benthic and nekton communities relative to unmodified sections of shoreline via habitat fragmentation (*Peterson & Lowe, 2009*), changes in nearshore erosion processes (*Bozek & Burdick, 2005*), increased depth of nearby waters (*Toft et al., 2013*), reduced species abundance and diversity (*Bilkovic et al., 2006*; *Bilkovic & Roggero, 2008*; *Kornis et al., 2017*; *Seitz et al., 2006*) at both local and landscape scales (*Isdell et al., 2015*), and prevention of landward migration of intertidal habitats (*Bilkovic, 2011*; *Titus et al., 2009*).

The ecological and social benefits of coastal wetlands (e.g., *Mitsch & Gosselink, 2015*) typically center around storm surge protection (*Spalding et al., 2014*; *Shephard & Grimes, 1983*), water quality enhancement (*Bilkovic et al., 2017a*; *Erwin, 2009*; *Nelson & Zavaleta, 2012*; *Zedler & Kercher, 2005*), habitat provision (*Angelini et al., 2015*; *Isdell, Bilkovic & Hershner, 2018*; *Rozas & Minello, 1998*), and carbon sequestration (*Davis et al., 2015*; *Mcleod et al., 2011*). Owing to the extensive ecosystem services provided by natural marshes and the unique challenges to protect coastal communities under changing conditions while supporting nearshore and intertidal ecosystems, nature-based shoreline protection is the preferred alternative to shoreline armoring where suitable. Nature-based shoreline protection, specifically living shorelines, provides a range of solutions that use or integrate natural features (e.g., planted marshes, shrubs, etc.) with engineered structures (e.g., a rock sill or concrete-based oyster reefs). The size and predominance of the engineered component are dependent on the physical setting of the shoreline (*Bilkovic et al., 2017b*), and areas with greater wave energy are likely to need more highly engineered structures.

Ecologically, living shorelines are generally viewed favorably and have been suggested by many (including the authors; e.g., *Bilkovic et al., 2017b*) as an alternative option to maintain ecosystem services while simultaneously protecting coastal property. Several studies have documented individual services and provided rate comparisons (*Bilkovic & Mitchell, 2013*; *Currin, Delano & Valdes-Weaver, 2008*; *Davis et al., 2015*; *Scyphers, Powers & Heck Jr, 2014*). Nevertheless, the absence of an assessment among multiple ecological criteria across an extensive geographic and project maturation range has resulted in varied implementation and piecemeal understanding of expected benefits. A comprehensive

comparison of ecosystem services provided by living shorelines and their nearby natural fringing marsh counterparts is urgently needed.

Here, we provide a synthesis of a multi-year study evaluating ecosystem function equivalency between between living shorelines and natural fringing marsh on the basis of nutrient cycling, primary production, and habitat provisioning for benthic and epifaunal invertebrates, nekton, and their predators (i.e, herons, and diamondback terrapins, *Malaclemys terrapin* (Schoepff) throughout Virginia's Chesapeake Bay. Soil components (C, N, P, and organic matter) in living shorelines are likely to lag behind natural marshes as a result of starting from clean sand fill during construction. Primary production (*Spartina alterniflora* (Loisel; cordgrass henceforth)) in living shorelines may initially lag behind natural marshes due to standard practices of sparse planting density (typically one stem/ft$^2$ (0.093 m$^2$) grid), but may rapidly increase as a result of fertilization during planting and wild recruitment. Benthic invertebrates will take time to colonize newly created living shorelines, but the length of time may vary considerably as a result of larval recruitment dynamics and dissimilar sediments. Nekton have been shown to quickly occupy a living shoreline within the first few years (*Gittman et al., 2016*), but nekton community characteristics, such as species abundance and biomass, may differ due to different structural features or marsh characteristics of living shorelines. Herons may benefit from the additional prey occurring in the rock sill of living shorelines, or use rock sill for hunting, roosting or preening. If nekton availability is similar between living shorelines and natural marshes, we predict no difference in heron use between marsh types. Terrapin may make use of a living shoreline immediately after construction for basking, while foraging will be dependent on prey availability. Together, these metrics provide a broad view of how living shorelines compare ecologically to natural marshes, and will provide valuable information to guide coastal resilience and adaptation planning.

## METHODS

### Study area

The study area was located along the East Coast of the United States in Virginia's portion of the Chesapeake Bay and its tributaries (Fig. 1). Salt marsh floral communities are dominated by cordgrass in the low marsh and *S. patens* ([Aiton] Muhl) in the high marsh, with *Juncus roemerianus* (Scheele) often occurring in the transition zone between low and high marsh. Overall, the Chesapeake Bay has approximately 1,861 km of armored shoreline, representing 8.5% of the total tidal shoreline (*Center for Coastal Resources Management, 2019*).

Sites were selected in pairs; one natural fringing marsh and one living shoreline in close proximity (<1 km apart) with similar physical settings for each site in a pair (Table 1). To standardize comparisons, all selected living shorelines were constructed using clean sand fill, cordgrass plantings, and rock sill. Pairs were distributed across a gradient from predominantly rural to predominately developed surroundings (see *Bilkovic et al. (2021)* for greater detail on site selection protocol). High marsh was sparse to non-existent at most of the NM sites, so comparisons were limited to the low marsh. The number of

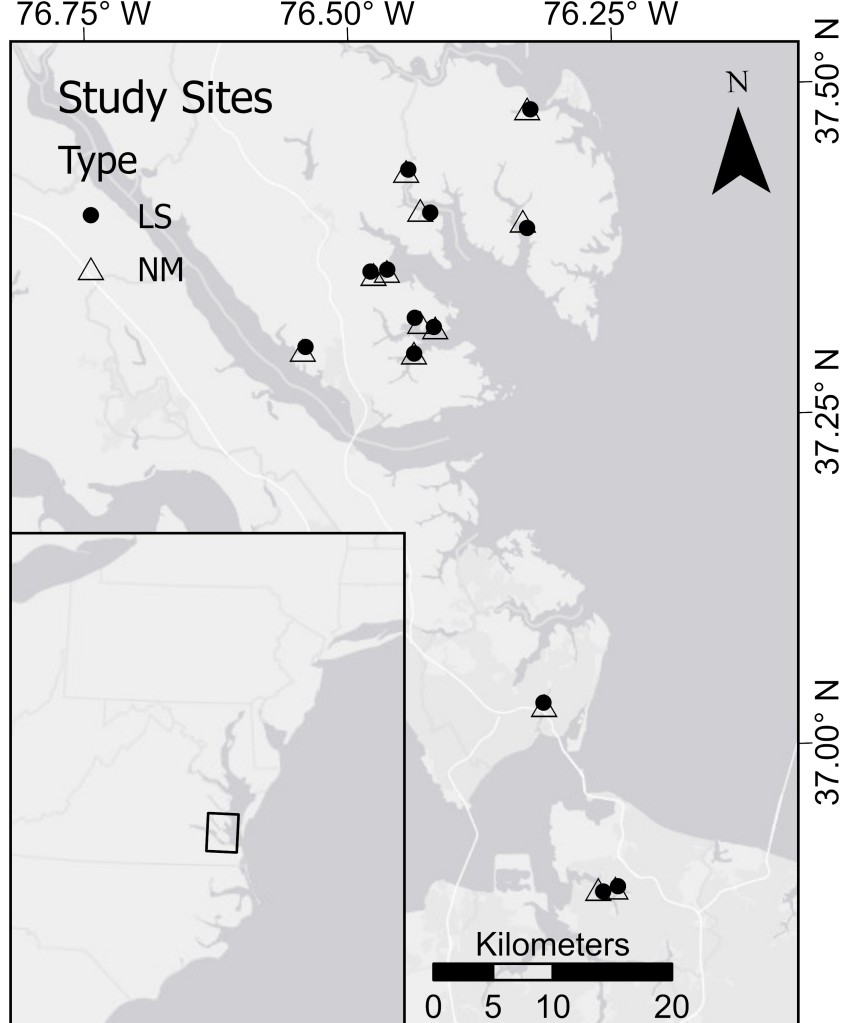

**Figure 1 Map of the study area.** There were a total of 13 pairs; living shorelines (LS) are marked by a black circle and natural marshes (NM) are marked by an open triangle.

pairs ($N = 13$) was constrained by the team's ability to obtain permissions for long-term, intensive ecological sampling on private property. Note, however, that to the best of our knowledge, 13 pairs is considerably larger than most studies to date, which typically include fewer than 5 living shoreline sites (*Smith et al., 2020*).

## Data collection

Data were collected during 2018 and 2019. Overall, sampling fell into the following five areas: soils ('Soils'), nekton ('Nekton'), benthic invertebrates and plants ('Benthic invertebrates and plants'), herons ('Herons'), and diamondback terrapin ('Terrapin'). We included 18 ecological measures that were proxies for ecological functions and indicative of ecosystem service provision. Detailed sampling methods for each of the five areas are provided below. Raw data are displayed in Fig. 2.

**Table 1  Living shoreline site characteristics.** In the table, the 'Age' column is the years since construction as of 2018. The 'Width' column is the mean (±SD) width (m) of the low marsh, perpendicular to the shoreline. The 'Length' column is the length (m) of the living shoreline, parallel to shoreline. The 'Tidal Amplitude' column is the mean tidal amplitude (m) for each site during July 2018. The 'Sill Height' column is the mean (±SD) height (cm) of the rock sill above mean high water.

| Pair | Age | Structure | Width | Length | Tidal amplitude | Sill height |
|---|---|---|---|---|---|---|
| 1 | 7 | Marsh Sill | 2.37 ± 0.31 | 55 | 0.78 | 15 ± 6 |
| 2 | 4 | Marsh Sill | 4.87 ± 0.34 | 25 | 0.72 | 9 ± 8 |
| 3 | 2 | Marsh Sill | 5.67 ± 1.17 | 29 | 0.72 | 0 ± 0 |
| 4 | 7 | Marsh Sill | 2.43 ± 0.50 | 42 | 0.81 | 0 ± 0 |
| 5 | 10 | Marsh Sill | 6.02 ± 1.76 | 34 | 0.81 | 7 ± 5 |
| 6 | 9 | Marsh Sill | 3.28 ± 0.57 | 63 | 0.78 | 6 ± 6 |
| 7 | 12 | Marsh Sill | 2.18 ± 0.46 | 55 | 0.78 | 0 ± 0 |
| 8 | 7 | Marsh Sill | 0.65 ± 1.24 | 61 | 0.38 | 1 ± 2 |
| 9 | 3 | Marsh Sill | 1.40 ± 0.91 | 30 | 0.78 | 23 ± 6 |
| 10 | 16 | Marsh Sill | 6.78 ± 2.89 | 77 | 0.72 | 0 ± 0 |
| 11 | 9 | Marsh Sill | 4.92 ± 0.66 | 59 | 0.78 | 11 ± 13 |
| 12 | 6 | Marsh Sill | 3.18 ± 1.25 | 39 | 0.78 | 2 ± 3 |
| 13 | 16 | Marsh Sill | 2.63 ± 0.80 | 72 | 0.75 | 0 ± 0 |

### Soils

*Methods.* During the 2018 growing season (May–August), soil cores to 20 cm were collected in the low marsh from three locations separated by at least 4 m horizontal distance along the shore. Detailed methods are available in *Chambers et al. (2021)*, but are provided here, briefly. Cores were collected from the low marsh, dominated by *S. alterniflora*, of each living shoreline and its paired, fringing natural marsh, then sectioned 0–5, 5–10 and 10–20 cm. For living shoreline marshes, plant roots had not yet penetrated deeper than 20 cm, so that depth was used for comparison with natural marshes. All core sections were oven dried at 60C and then bulk density was determined gravimetrically. From dried sub-samples of cores homogenized at each depth, organic content was calculated from weight loss after ashing for 4 h at 450C. Total carbon (C) and nitrogen (N) were determined using a Perkin-Elmer 2400 elemental analyzer, and total phosphorus (P) was determined using an ashing/acid hydrolysis method (*Chambers & Fourqurean, 1991*).

*Analyses.* Soil nutrient standing stocks to 20 cm were calculated and presented as weight percentages using the weighted mean of each nutrient for each core at each site. Site-level means and standard deviations were calculated as the mean value of each metric across all three cores down to 20 cm.

### Nekton

*Methods.* Living shoreline and paired natural marsh sites were sampled once during summers (mid-June to early-August) in 2018 and 2019, when marsh fish abundance and diversity is greatest in Virginia (*Bilkovic et al., 2012*). Paired sites were sampled concurrently to ensure similar environmental conditions. None of the sites were in close proximity to other potential structural nursery habitats (e.g., persistent seagrass beds) to minimize

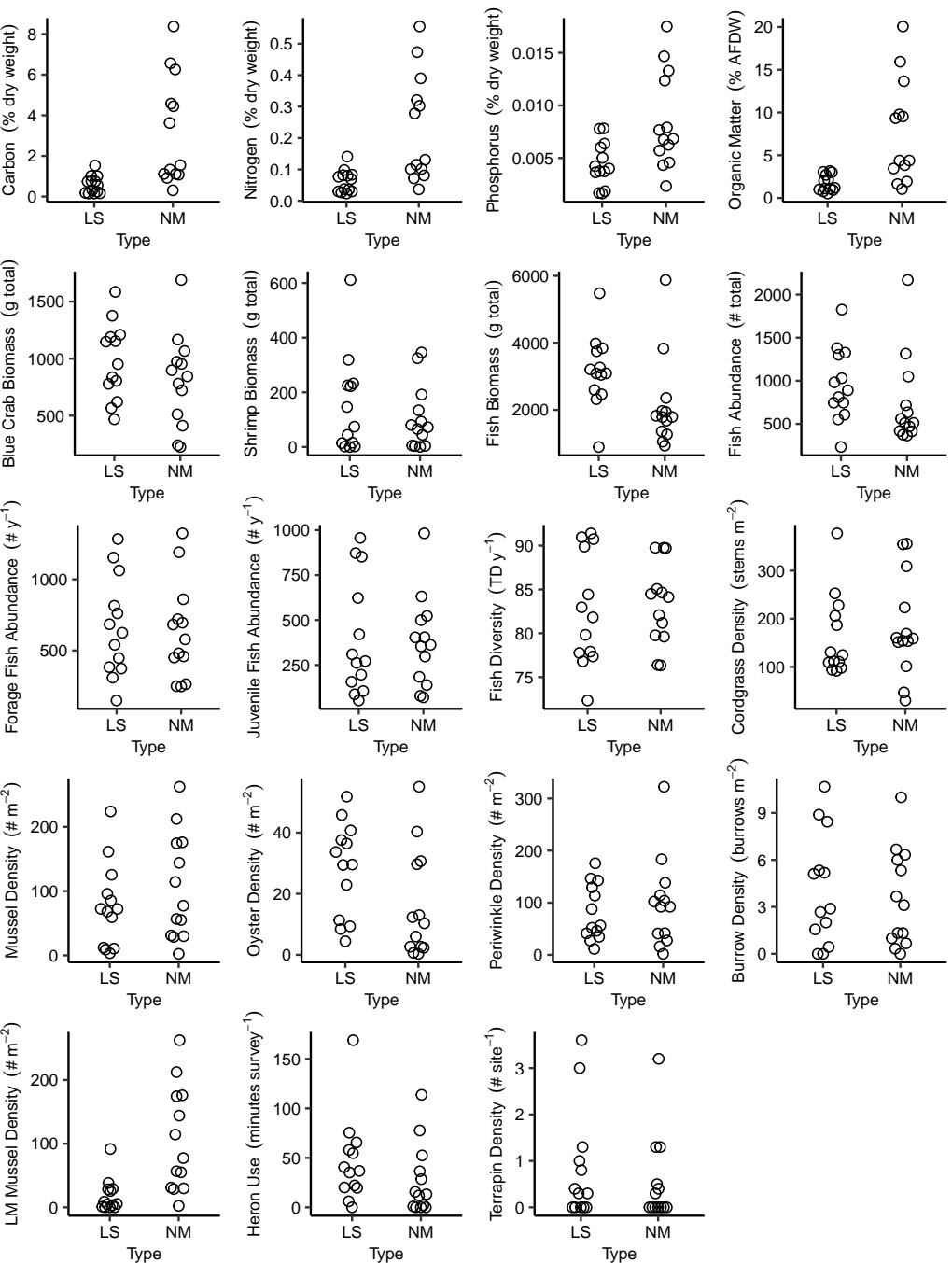

**Figure 2** **Bee swarm plots of the raw data for each metric grouped by site type.** Each plot displays site-level data for a given metric, grouped by site type (LS = living shoreline, NM = natural marsh). The panel LM Mussel Density is the data for the mussels that are only in the low marsh.

potential confounding factors. We used multiple gear types to sample subhabitats of the site, including marsh edge (fyke net) and marsh surface (minnow traps). At each site we fished 2 fyke nets and set 10 minnow traps. All nekton were collected following William & Mary IACUC Protocol #2019-05-21-13670.

Nekton use of vegetated marsh surface was assessed by setting replicate ($n = 10$) minnow traps at each site. Traps were set at high tide and retrieved once the funnel openings were above water for an average of $2.7 \pm 1.4$ hr (SD) soak time. All traps were set in the low marsh habitat with five traps set at the water edge of the marsh (within the first meter) and five traps set closer to the low marsh/high marsh boundary ($\sim 2$ m from the marsh-water edge). Traps were haphazardly placed at least 1 m apart.

At each site, two fyke nets were set at high tide and retrieved once the tide had dropped below the surface elevation of the low marsh. Each net fished for an average of 4 hr $\pm$ 40 min (SD). Fyke nets were placed at the sill gaps or ends of the living shoreline sites and randomly along the edge of natural marsh sites. Fyke net openings were set at the same distance from marsh edge ($\sim 1$ m, depending on sill location relative to the marsh edge). The fyke nets consisted of a $0.9 \times 0.9 \times 3.0$ m compartmentalized, 3.175 mm mesh bag with $0.9 \times 5.2$ m wings that stretched out from the bag (set for a total mouth width of 8 m) into the marsh to the high-water line.

All captured fish were sorted and counted by species. For each species and sampling effort (e.g., first fyke net), fish were individually measured (total length) and total weight by species was recorded. For highly abundant species, a subsample of 25 was measured and weighed. All blue crabs (*Callinectes sapidus* [Rathbun]) were individually measured (carapace width, cw, mm), weighed (g), and sexed. Grass shrimp (*Palaemonetes* spp.) and white shrimp (*Panaeus setiferus* [L.]) were counted and total biomass was recorded for each sampling method.

*Analyses.* We summed the biomass (g) of fish, blue crabs, and shrimp separately that were captured in the intertidal marsh (fyke and minnow pots) across 2018 and 2019 for each site. For fish, we calculated forage base abundance and juvenile abundance averaged across 2018 and 2019. The forage base was defined as fish that are primary and secondary consumers and are often consumed by carnivorous fish (*Ihde et al., 2015*). Using this subset, we categorized if the individuals were young-of-year using established literature values (see Table S1 in *Guthrie et al., accepted*) for full documentation of species-specific size thresholds and citations therein).

We calculated site-specific nekton (fish and crustacean) diversity using Taxonomic Distinctness (PRIMER v7), then averaged across 2018 and 2019. Annual site-level averages for forage and juvenile fish abundance and fish diversity were averaged to get the across-year site-level means used in this analysis.

### Benthic invertebrates and plants

*Methods.* Invertebrate data were collected near low tide during the fall of 2018. At each site, six transects were placed perpendicular to the shoreline, spaced at least 5 m apart, and divided into one (natural marsh) or two (living shoreline) sampling zones, rock sill (living

shoreline only) and low marsh (both). Each zone was sampled using 0.25 m² quadrats placed to the right side (when facing inland) of the transect. At each transect along the rock sill, the quadrat was placed with the center of the quadrat spanning the mean water line, which was approximately the same elevation as the marsh surface at the front (waterward) edge, for a total of six samples per site. In the low marsh of each transect, one quadrat was placed at the leading (water) edge and one at 1 m inland from that point for a total of 12 samples in the low marsh at each site. Within each quadrat, the number of individuals of each identifiable invertebrate species and cordgrass stem counts were recorded. All visible ribbed mussel (*Geukensia demissa* [Dillwyn]), oyster (*Crassostrea virginica* [Gmelin]), and periwinkles (*Litoraria irrorata* [Say]) that occurred anywhere within the aboveground space of the quadrat were counted. The number of burrowing crab (combined members of the family Occypodidae and *Sesarma reticulatum* [Say]) burrows were also recorded.

*Analyses.* For this synthesis, we extracted the mean mussel, oyster, periwinkle, crab burrow, and cordgrass densities. Bivalve densities were averaged across all low-marsh quadrats ($N = 12$) at NM sites and low-marsh plus sill quadrats ($N = 18$) at LS sites, while periwinkle, crab burrow, and cordgrass densities were averaged across all low-marsh quadrats only, and a within-site standard deviation (SD) was calculated. A follow-up analysis focused solely on the low-marsh quadrats (i.e., without the sill) at both LS and NM sites to assess the relative role of the sill structure in comparisons of functional equivalence for bivalves.

### Herons

*Methods.* We recorded heron activities remotely using cameras surveying each site between one to three times from May until August in 2018 and 2019, or a total of three to six times across both years. We ensured equal survey effort within each living shoreline-natural marsh pair and year. For each survey, we placed between 3–6 cameras on the rock sill of a living shoreline or at the edges of the paired natural marsh to ensure complete sampling of each site's extent. Cameras were placed the night before the day of the survey to avoid disturbing herons and were affixed to a tripod, approximately 1 m above the high-tide level. We programmed cameras to record 4 30-minute segments per survey and site near the expected peak activity times for herons (i.e., sunrise and sunset) as well as high tide and low tide (*Burger, Niles & Clark, 1997*). Because light levels were too low at sunrise or sunset, recordings were timed an hour after sunrise and before sunset to ensure herons would be visible in the video. In 2018, we employed Raspberry Pi cameras (Naturebytes Wildlife Cam Kit; https://shop.naturebytes.org/product/naturebytes-wildlife-cam-kit/). However, camera performance was negatively affected by high ambient temperatures, leading to fewer and lower quality segments being recorded. Thus, during 2019, we used GoPro Hero 5 (GoPro, Inc., San Mateo, California, USA) cameras with a BlinkX Time Lapse Controller (CamDo Solutions, Vancouver, BC, Canada) and DryX Weatherproof Enclosure (CamDo Solutions, Vancouver, BC, Canada). Bird sampling was done in accordance with William & Mary IACUC protocol #2016-06-14-11270.

We estimate heron site use by scanning each 30-min segment for presence of herons. Once a heron was detected, we measured site use to the nearest second with a stopwatch. If a given heron was detected on two or more cameras simultaneously, we only recorded time from one camera, usually the one with the best view of the heron.

*Analyses.* We used adjusted total observation time by sampling effort for herons utilizing either a living shoreline or natural marsh. Adjusted total observation time was calculated by dividing total observation time, aggregated across 2018 and 2019, divided by the total recording time of all cameras placed at a given site and year. Heron species included Great Blue Heron (*Ardea herodias* [L.]), Great Egret (*Ardea alba* [L.]), Green Heron (*Butorides virescens* [L.]), and Yellow-crowned Heron (*Nyctanassa violacea* [L.]).

### Terrapin

*Methods.* We used visual surveys to estimate diamondback terrapin densities. Surveys were completed between one to three times for 30 min between mid-May and August (comprising the terrapin nesting season) in 2018 and 2019, or a total of 3–6 times per site across both years. For each sampling occasion, observers noted factors that could influence terrapin detection, such as day of year (Julian date), the starting time of a survey, and cloud cover as quantiles (0, 25, 50, 75, or 100%). We also measured at the beginning, middle and end of each survey wind speed and temperature with a hand-held weather station (Kestrel 2000 Wind Meter). Once a terrapin was detected, we estimated the distance between an observer and a given individual using 8× monocular laserrangefinder (Zeiss Victory PRF; Germany) and noted size (small vs. large) on the basis of the head and coloration (black, black-white, and white). We used coloration to reduce sampling the same individual multiple times. This sampling and data structure enabled us to estimate terrapin density adjusted for imperfect detection. Terrapin data were collected in accordance with William & Mary IACUC protocol #2019-03-29-13573. We modeled the detection process on the basis of covariates collected during each sampling occasion.

*Analyses.* Terrapin use of living shorelines and natural marshes was included as the head count (unique individuals) per unit effort (hours of observation per site). We estimated total head counts across both years within the effective radius surveyed, which is the distance at which an observer is as likely to miss a terrapin within that distance as to detect an individual beyond it (*Buckland et al., 2001*). We estimated effective area surveyed by first fitting two key functions, including three series expansions for each key function in program Distance (*Buckland et al., 2001*; *Thomas et al., 2010*). We omitted 5% of the farthest observations as recommended by *Buckland et al. (2001)*. Once the best detection function was selected, we evaluated model fit on the basis of the five aforementioned environmental covariates. We estimated effective radius surveyed from the model with the lowest Akaike's Information Criterion (AIC, *Burnham & Anderson, 2002*) value and a non-significant Goodness-of-Fit test (*Buckland et al., 2001*; *Thomas et al., 2010*).

## Data synthesis

Metrics were compared, contrasted, and combined using a $Z$-score approach. For each metric of each pair, a $Z$-score was calculated using either a local (within-pair) SD (Formula (1)) or a regional (among-pair) SD (Formula (2)):

$$\sigma_{L_i} = \sqrt{\frac{\sigma^2_{LS_i} + \sigma^2_{NM_i}}{2}} \tag{1}$$

$$\sigma_R = \sqrt{\frac{\sigma^2_{LS} + \sigma^2_{NM}}{2}} \tag{2}$$

where $\sigma^2_{LS_i}$ is the SD of the metric at living shoreline of the $i$th pair and $\sigma^2_{NM_i}$ is the SD of the metric at natural marsh of the $i$th pair. Metrics derived from sampling procedures that involved identical effort and replication within a site (i.e., soils, invertebrates, and plants) were eligible for the local SD while all the other metrics relied on the regional SD. Although the nekton were collected using standardized procedures, combining catch from different gear types precluded a local estimate of the standard deviation. For each metric, the natural marsh value was subtracted from the living shoreline value, with positive $Z$-scores indicating that living shorelines provided a higher level of function than the natural marsh and negative scores indicating a lower level of function.

The mean $Z$-score was calculated across all metrics for each pair to yield a net functional equivalence score. We considered a metric to be functionally equivalent between the living shoreline and the natural marsh if $|Z\text{-score}| < 1$. Note that all metrics are structural in nature, which serve as proxies for ecosystem functions rather than explicit measurements. We compared the net functional equivalence score to the living shoreline age (years since construction as of 2018) using Bayesian non-linear regression implemented in the R package "brms" (*Bürkner, 2018*). We fit a logarithmic curve consistent with our assumption that there would be an asymptotic relationship where living shorelines would eventually reach and maintain functional equivalence with natural marshes instead of outperforming them to an indefinite amount after reaching equivalence. While a logarithmic curve is not a true asymptotic relationship, it does closely approximate one in a small predictor space while reducing the complexity of the model–an essential consideration with a sample size of 13. The model was specified as:

$$y_i \sim Normal(\mu_i, \sigma)$$
$$\mu_i = \beta_0 + \beta_1 ln(x_i)$$
$$\beta_0 \sim Normal(0, 1)$$
$$\beta_1 \sim Normal(0.25, 0.75)$$
$$\sigma \sim Gamma(1, 1)$$

where $y_i$ represents the net functional equivalence score for pair $i$, and $x_i$ is the age of the living shoreline of pair $i$. Parameter $\beta_0$ is the intercept and $\beta_1$ is the estimate of the slope

**Table 2  Z-scores for each ecological metric.** The mean $Z$-score for each ecological metric was calculated across all 13 living shoreline/natural marsh pairs.

| Metric | Score |
| --- | --- |
| Mussels | −0.20 |
| Mussels (LM Only) | −0.80 |
| Oysters | 0.28 |
| Periwinkles | −0.12 |
| Burrows | 0.02 |
| Cordgrass | −0.14 |
| Organic Matter | −1.86 |
| Carbon | −2.61 |
| Nitrogen | −2.60 |
| Phosphorus | −1.76 |
| Fish Biomass | 0.85 |
| Crab Biomass | 0.46 |
| Shrimp Biomass | 0.28 |
| Fish Abundance | 0.48 |
| Juvenile Fish Abundance | 0.06 |
| Forage Fish Abundance | 0.09 |
| Fish Diversity | −0.12 |
| Heron Use | 0.49 |
| Terrapin | 0.27 |

for age. Weakly informative priors were used for the intercept ($\beta_0$) and standard deviation ($\sigma$), but a modestly informative prior was set to reflect expectations of a slightly positive effect of age ($\beta_1$) given previously published literature (*Onorevole, Thompson & Piehler, 2018*; *Bilkovic et al., 2021*; *Chambers et al., 2021*) as well as our original hypothesis that living shorelines would become more similar to natural marshes over time. The model was run with four chains for 50,000 iterations and 5,000 warm-up samples, and compared to the null model using leave-one-out cross-validated information criterion (LOOIC; *Vehtari, Gelman & Gabry, 2017*).

## RESULTS

Overall, all services except for the soil characteristics had a mean absolute $Z$-score of <1 SD (Table 2). The mean of all $Z$-scores was −0.34 ± 1.08 (mean ± SD; Fig. 3).

### Soil

Among the four metrics, percents organic matter, C, N, and P, all mean $Z$-scores were <−1 indicating that these were the only metrics for which there was a considerable lag in ecosystem development of the living shoreline in comparison to the natural marsh (Fig. 3). Both C and N scored the lowest with the values of −2.61 and −2.60, respectively. All living shoreline soils had a lower mean percent C compared to their natural marsh pair, but the strongly negative overall mean value was largely driven by two pairs (2 and 9) which received $Z$-scores of −10.23 and −6.52 respectively. Results were similar for percent

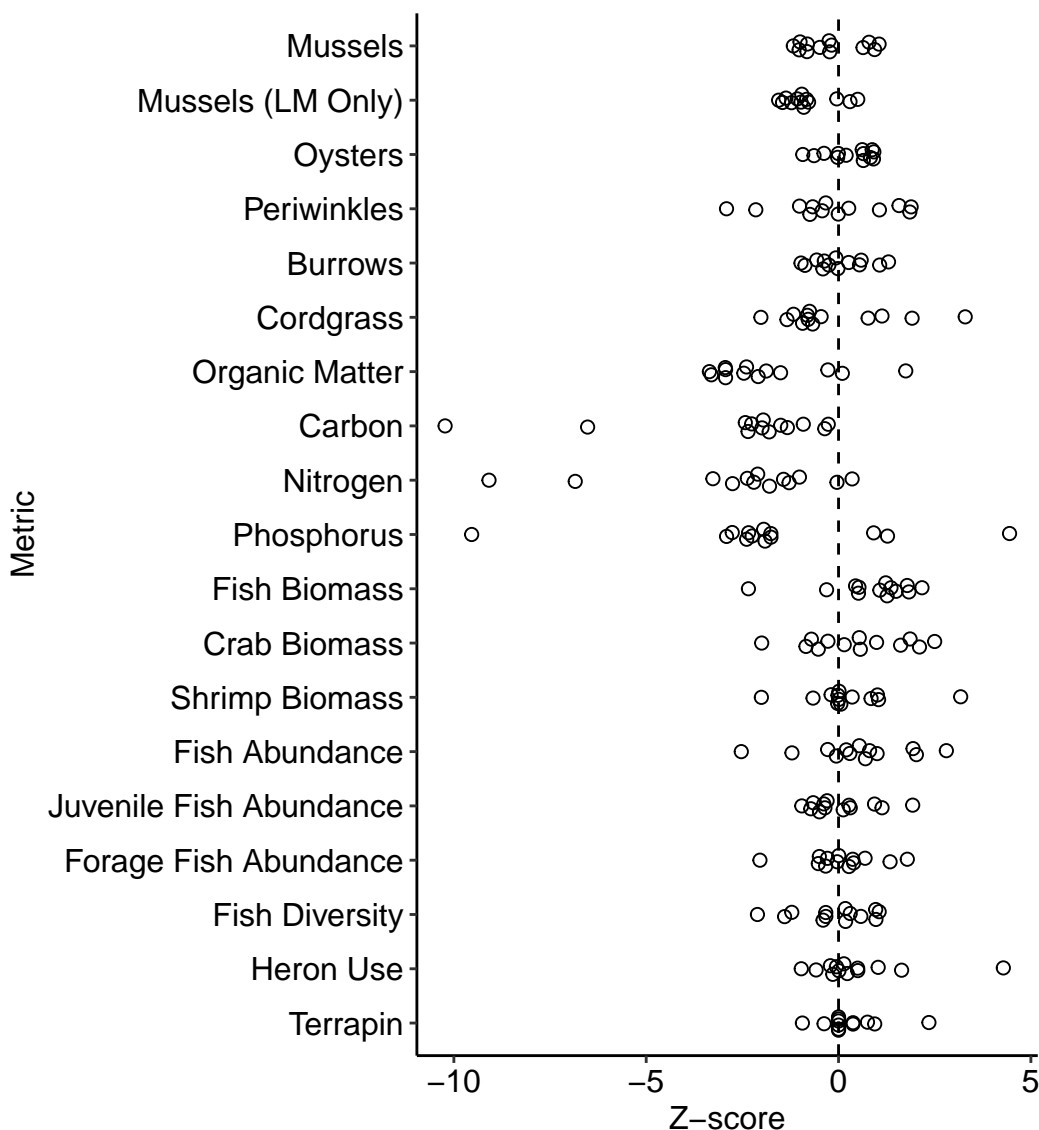

**Figure 3** **Pair-level $Z$-scores for ecological metrics.** Most $Z$-scores are tightly clustered around zero (dashed vertical line) except for the soils which display a generally negative grouping.

N, but the magnitudes were slightly lower: one pair was positive, and pairs 2 and 9 received scores of −9.09 and −6.84, respectively.

## Nekton

Among the nekton metrics, all $Z$-scores were within one SD of zero (Fig. 3), and all but fish diversity were slightly positive indicating that living shorelines performed as well as or better than natural marshes, though not by a large amount. Individual pairs had large differences between the living shoreline and natural marsh for some metrics (Table S1; $|Z| \geq 2$), but those differences were not consistently in favor of either the living shoreline or natural marsh and approached parity at the regional scale.

### Invertebrates and plants

Mussel, oyster, periwinkle, burrowing crab, and cordgrass densities were all similar between living shorelines and natural marshes, scoring −0.20, 0.28, −0.12, 0.02, and −0.14 respectively. Among pairs, most mussel and burrowing crab, and all oyster $Z$-scores were within one SD of zero (Fig. 3). Periwinkles and cordgrass had a much larger range among pairs (Fig. 3). Although $Z$-scores at a majority of pairs for both periwinkles and cordgrass (8/13 and 9/13 pairs, respectively) were negative, the median values were −0.33 and −0.75, respectively, which still indicates similar functional equivalence overall.

When we only considered the mussels that occurred within the low marsh at living shorelines and excluded those that occurred on the sill, the overall $Z$-score shifted to be slightly more negative (−0.8) with a similar range (−1.56 to 0.50) but still relatively low overall indicating similar functional equivalence. The absolute differences were much larger ($>250$ mussels $\cdot$ m$^{-2}$ in the natural marsh of one pair; Fig. 2), but the overall $Z$-scores were still low as a result of the high local variance in mussel densities among quadrats at each site. Oysters occurred almost exclusively on the sills at living shoreline sites. Only 3/13 sites had any oysters, and those that did had densities $\leq 1$ oyster m$^{-2}$. In contrast, every natural marsh had at least some oysters (0.7–55.0 oysters m$^{-2}$).

### Herons

Herons had a low overall $Z$-score (0.49) indicating overall similar use at both living shorelines and natural marshes. Scores were strongly right-skewed for herons (Table S1). We observed herons at 22 of the 26 sites but did not detect any herons at one living shoreline nor at three natural marshes in both years. Of those sites used by herons, total observation time ranged between 1.0–168.7 min. In total, we observed herons for 4.5 hrs and 2.9 h at living shorelines and natural marshes sites, respectively.

### Terrapin

Observed terrapin use was equivalent between living shorelines and their natural fringing marsh pair ($Z = 0.27$). The effective radius surveyed was 43.0 m (95% CI: [26.6 m–69.5 m]), estimated on the basis of the hazard-rate detection function with wind speed included as a covariate. Across both years, we detected 178 terrapins, of which 169 detections were used to estimate effective radius surveyed. Within the effective radius surveyed, we detected 41 terrapins, with more detections in living shoreline sites ($n = 24$) than the natural marsh sites ($n = 17$).

### Age

We did not find strong evidence to support the hypothesis that net functional equivalence score would increase over time (Fig. 4; $\beta_{age} = 0.33$, −0.25 to 0.88; mean, 95% credible interval; Fig. S2). The logarithmic growth model (LOOIC = 33.5) was within 2 $\Delta$LOOIC of the null model (LOOIC = 34.8), indicating minimal difference between the two models. Additional samples may be able to detect a difference given that a small percentage (12.5%) of the distribution overlapped zero. The same model with values of mussels only in the low marsh substituted for mussels in both the marsh and the sill performed similarly. The model estimated that $\beta_{age} = 0.33$ [−0.26, 0.87], with 12.7% of the posterior distribution

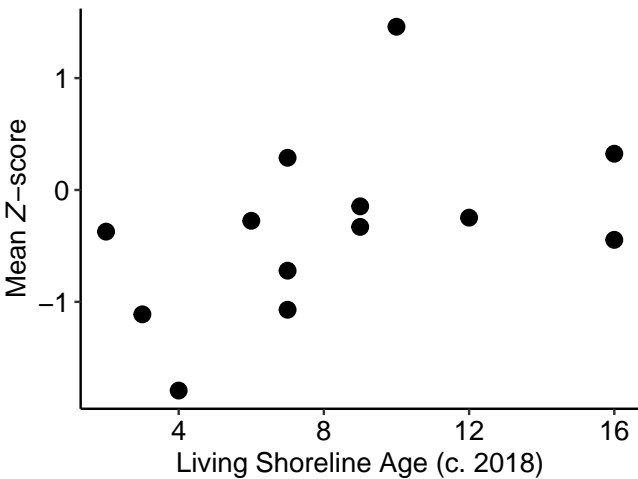

**Figure 4  Net functional equivalence score vs living shoreline age.** There was no detectable effect of the time since living shoreline construction on the net functional equivalence score across the 13 pairs.

overlapping zero (Fig. S4) and <2 ΔLOOIC between the null and the growth model. Posterior predictive checks for both models indicated that the mean and variance of the data were adequately described by the assumed normal distribution (Fig. S1 and Fig. S3). Scatter plots of $Z$-score vs. age for each metric are available in Fig. S5.

## DISCUSSION

Across nearly all metrics assessed in this study, living shorelines were not functionally different from natural fringing marshes. Nekton, invertebrates, plants, herons, and terrapin all occurred in living shorelines at levels similar to or greater than their natural fringing marsh counterparts. Owing to their construction using clean sand fill with high bulk density and low organic content, living shorelines received consistently lower scores than their natural marsh counterparts. With rapid plant growth and organic accumulation in living shorelines, however, even soil composition is expected to achieve equivalence over time (*Chambers et al., 2021*). These findings provide encouraging support that living shorelines are capable of providing the same ecosystem services that natural fringing marshes have provided historically. Living shorelines, specifically marsh sills, incorporate an engineered structure to reduce erosion and provide longer-term stability of the front edge of the marsh. This long-term stability coupled with net functional equivalence to natural fringing marshes suggests that living shorelines should be able to contribute to increased ecological resilience of a shorescape (defined here as the aquatic-terrestrial ecotone along a reach of shoreline, akin to landscape and seascape) to sea level rise.

Among the nekton metrics, biomass (fish, blue crabs, and shrimp), fish abundance (all fish, juvenile fish, and forage fish), and taxonomic distinctness were equivalent between the living shorelines and their reference natural fringing marshes. This is a clear indication that nekton use of the created marshes of living shorelines is comparable to natural marshes, given that our assessment targeted the fish caught on the marsh surfaces and edges.

While there are nuances among different species (*Guthrie et al., accepted*), the overall trends suggest that across taxa and time, living shorelines do provide quality habitat for ecologically and economically important species. The similar abundances of juvenile fish, forage fish, and blue crabs in living shorelines relative to natural marshes also suggests that these created habitats are able to serve as important nursery habitat, refuge, and foraging opportunities for many species (*Bilkovic et al., 2020*).

One particularly important nuance of our findings is the role of the sill structure in achieving functional equivalence for bivalves. Ribbed mussel recruitment and survival in the low marsh of living shorelines lags well behind what we observe in nearby natural fringing marshes (*Bilkovic et al., 2021*), likely as a result of a challenging post-settlement environment. There is something of a quandary in which the lack of adult conspecifics increases predation and desiccation risk to new recruits. These effects are exacerbated by low soil-moisture content (as a result of using clean sand with a low organic matter content) and an immature root mat/lack of peat to help secure mussels in place, which means fewer juveniles survive to adulthood to facilitate recruitment (*Nielsen & Franz, 1995*). However, the sill structure seems to provide the protected nooks and crannies that allow rapid establishment of ribbed mussel populations. When considered as a whole (low marsh *and* sill), living shorelines support similar numbers of ribbed mussels as their natural fringing marsh counterparts. This means that we would expect similar levels of filtration, with its implications for water quality, at living shorelines. However, research has found that denitrification rates are highest when the mutualistic relationship between cordgrass and ribbed mussels is intact (*Bilkovic et al., 2017a*). Decoupling the ribbed mussels from the cordgrass by primarily supporting the mussels on the sill may have implications for estimating the N removal potential of living shorelines relative to natural fringing marshes. The absence of a healthy ribbed mussel population in the living shoreline marshes may also contribute to the lagging maturation of the soils. Ribbed mussels are capable of contributing organic matter to the soil directly via biodeposition (*Jordan & Valiela, 1982*; *Smith & Frey, 1985*) and indirectly by fertilizing cordgrass, which facilitates both above- and below-ground growth (*Bertness, 1984*). The near total absence of oysters in the low marsh of living shorelines further highlights the importance of considering the role of the sill structure for bivalves at living shorelines.

This is the first study to evaluate ecological function of living shorelines for herons. Heron use did not differ between living shorelines and natural marshes, indicating that living shorelines provide additional habitat for these species. As discussed above, prey base for herons, which includes invertebrates and fish (*Davis Jr & Kushlan, 2020*; *McCrimmon Jr et al., 2020*; *Vennesland & Butler, 2020*; *Watts, 2020*), was equal to or greater at living shorelines compared to natural marshes. Therefore, living shorelines provide herons with additional habitat for foraging. Because behavioral observations were limited to the daytime, nighttime use of living shorelines and natural marshes by herons warrants further evaluation. In addition, living shorelines use by shorebirds also needs to be evaluated.

Diamondback terrapin, a state-listed species of special concern (i.e., near-threatened) in Virginia, were found in similar abundance at both living shoreline sites and natural fringing marshes. While the survey data were based on headcounts of terrapin immediately

offshore (within 43 m of the observer), video footage collected for other components of the larger study also observed terrapin basking on the rock sill and moving around in both the created marsh and the natural marsh (Fig. S6 and Fig. S7). Two terrapin were also caught in the fyke nets at living shorelines as part of the nekton sampling. Given that terrapin diet items (blue crabs, periwinkles, and small fish; (*Tulipani, 2013*) were present in similar numbers at living shorelines and natural fringing marshes, terrapin are likely foraging at living shorelines. The rock sills may also provide additional or better basking habitat, given that the tops of the sills are often out of the water, even at high tide. Terrapin, which are strongly tied to both marsh and structure at large (250–1,000 m) spatial scales (*Isdell et al., 2015*), may also benefit in the long term from the stabilized marsh sills when many areas are likely to lose natural fringing marshes as a result of sea level rise (*Isdell, Bilkovic & Hershner, 2020*).

Evidence presented here and elsewhere (*Chambers et al., 2021*; *Currin, Delano & Valdes-Weaver, 2008*; *Davis et al., 2015*) also indicates that soils, the only metrics that did not achieve overall equivalence, are able to reach similar levels to natural reference marshes over time. Soil organic matter and nutrients in plant root zone generally accrue with living shoreline age, but the accumulation rates are non-linear. With rapid vegetation establishment (*Bilkovic et al., 2021*; *Currin, Delano & Valdes-Weaver, 2008*), however, and longer-term carbon sequestration and nitrogen and phosphorus accumulation over timescales measured in decades (*Chambers et al., 2021*; *Davis et al., 2015*), living shorelines appear to be on a trajectory to approach soil equivalence with natural reference marshes.

When considering all of the assessed metrics for each of our pairs, we did not find strong evidence that older living shorelines were more similar to their reference natural fringing marshes than younger living shorelines (at least for living shorelines $\geq$ 2 year old). This may, at first, seem counter-intuitive given that there is a considerable body of work demonstrating that several of the metrics that we measured increase over time after construction of a living shoreline (*Gittman et al., 2018*; *Chambers et al., 2021*). However, while some metrics are likely to increase over time (e.g., soil nutrients), others may be able to take advantage of the site immediately (e.g., terrapin basking on the rocks). Nekton may be able to take full advantage of the site as soon as the cordgrass has grown and their prey begin to colonize the sand fill. Cordgrass may reach high densities within the first two years after planting, or shortly thereafter [pers. obs.]. Benthic epifauna may also quickly colonize new sites as a result of plantonic larvae and favorable conditions. Once the majority of the metrics in an aggregated analysis achieve equivalence between living shorelines and natural marshes, age effects will be harder to detect. The potentially quick ramp-up for most of our measured metrics combined with only a few sites $\leq$ 5 years old, could explain the lack of an observable effect of age on the overall $Z$-score. With this, considerable differences were observed between living shorelines and their paired natural fringing marsh (Table S1) at both a site-level and for individual metrics, indicating that site-specific differences (e.g., living shoreline design, geographic setting, and general ecological health of the surrounding shorescape) may be more important than the age of the living shoreline, ultimately speeding up or slowing down the time required to reach functional equivalence. It is possible, however, that poorly designed or sited living shorelines may never reach

functional equivalence. Our work suggests that living shorelines are capable of providing similar levels of ecosystem services along shorescapes; not that any specific living shoreline project *will* provide those services without proper design and spatial context.

Given our findings and the projected acceleration of SLR and wetland loss (*Boon et al., 2018*; *Mitchell et al., 2017*), living shorelines may be able to offset the ecosystem services lost from the degradation of natural fringing marshes. This would require considerable expansion and implementation of living shorelines relative to current levels. For example, if living shorelines were implemented along all stretches of shoreline where they were both suitable and where some form of shoreline protection is warranted in Virginia's portion of the Chesapeake Bay (Karinna Nunez, 2021, unpublished data), there would be an additional 10,714 km of marshy shoreline, representing >75% of all shoreline where some form of erosion control is needed.

## CONCLUSIONS

This work supports the underlying assumption that living shorelines enhance intertidal ecosystem resilience to climate change and provide comparable ecosystem functions as natural fringing marshes. While the geographic scope of our work was restricted to Chesapeake Bay, Virginia, USA, the ecological processes and anthropogenic pressures in our study area are common along the US Atlantic seaboard as well as other temperate regions wherever salt marshes occur. If living shorelines are constructed according to best design practices (*Bilkovic & Mitchell, 2017*; *Bilkovic et al., 2021*), they can provide functional equivalence to natural fringe marshes for most of the 17 ecological metrics that we examined. Living shorelines are constructed in a way that reduces erosion and allows for landward migration with SLR, thereby making them a resilient alternative to shoreline armoring while maintaining functional equivalence to natural fringing marshes.

## ACKNOWLEDGEMENTS

We would like to acknowledge the field and data entry help of Kory Angstadt, Dave Stanhope, Adrianna Gorsky, Robert Galvin, Samuel Mason, and Jesse Smyth, as well as numerous undergraduates from William & Mary and other volunteers assisted in the field work. This study would not have been possible without the gracious homeowners who granted us permission to intensively sample their shorelines. We would also like to thank David Johnson and two anonymous reviewers for their valuable feedback that improved the manuscript.

### Funding

This material is based upon work supported by the National Science Foundation under Grant Number 1600131. The funders had no role in study design, data collection and analysis, decision to publish, or preparation of the manuscript.

## Grant Disclosures

The following grant information was disclosed by the authors:
National Science Foundation under Grant Number: 1600131.

## Competing Interests

The authors declare there are no competing interests.

## Author Contributions

- Robert E. Isdell conceived and designed the experiments, performed the experiments, analyzed the data, prepared figures and/or tables, authored or reviewed drafts of the paper, and approved the final draft.
- Donna Marie Bilkovic, Amanda G. Guthrie, Molly M. Mitchell, Randolph M. Chambers, Matthias Leu conceived and designed the experiments, performed the experiments, analyzed the data, authored or reviewed drafts of the paper, and approved the final draft.
- Carl Hershner conceived and designed the experiments, authored or reviewed drafts of the paper, wrote the original grant and conceived the overarching study, and approved the final draft.

## Animal Ethics

The following information was supplied relating to ethical approvals (i.e., approving body and any reference numbers):

William & Mary provided full approval for this research (nekton: IACUC Protocol # 2019-05-21-13670; birds: #2016-06-14-11270; terrapin: #2019-03-29-13573).

## Data Availability

The R scripts, figures, and data used in this analysis are available at OSF: Isdell, Robert E, Donna M Bilkovic, Amanda Guthrie, Molly M. Mitchell, Randolph Chambers, Matthias Leu, and Carlton Hershner. 2021. "Living Shorelines Achieve Functional Equivalence to Natural Fringe Marshes across Multiple Ecological Metrics." OSF. June 28. osf.io/7vzdp.

## Supplemental Information

Supplemental information for this article can be found online at http://dx.doi.org/10.7717/peerj.11815#supplemental-information.

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
