# Peer review of "Living shorelines achieve functional equivalence to natural fringe marshes across multiple ecological metrics"

_PeerJ, doi:10.7717/peerj.11815_

## Round 0.1 · original submission · Major Revisions

Dear authors,
Congratulations on your interesting and valuable paper. We have received two thorough reviews of you paper which will likely improve the quality of your work. Both reviewers are somewhat concerned with the spatial and temporal replication, and how your study design may influence or prevent some interpretations and conclusions. Reviewer #1 also recommends presenting raw data, which would indeed allow future use of your work by others. I am looking forward in receiving your revised manuscript.

Best regards

Reviewer 1 ·

Basic reporting

The manuscript was clear, interesting, and well referenced.

Experimental design

The experimental design was clear and the methods were described in sufficient detail (as long as the comments below are addressed).

Validity of the findings

The findings are valuable and conclusions are well stated.

Additional comments

Summary:
I enjoyed reading this manuscript and I think it will make a very valuable contribution to the field. I have mostly minor comments. My only major comment is that I think the paper would greatly benefit from a figure(s) showing the raw data. The authors only present the z-scores and occasionally the range in z-scores, but they don’t provide any indication of the absolute values of fish biomass, for instance, or the variation across sites. This paper will be more broadly applicable and useful for other researchers if the authors can also present the average biomass, nutrients, abundance, etc. across all of the sites.

Minor comments:
Introduction:
• Lines 34-35: Change to: “Losses are likely to be exacerbated where these conditions overlap with extensive watershed development”
• Line 39: Change “Whereas” to “While”
• Lines 55-58: I don’t agree with the use of the term nature-based features. Hardened shorelines can contain nature-based features (i.e. flower pots on seawalls for added habitat value), but the authors present these as distinct shoreline alternatives. I think the term nature-based infrastructure is more commonly and generally used and would be more appropriate here. But, I acknowledge that there is a diversity of terminology used in the field and not always consistently, so perhaps this is personal preference. The authors don’t have to respond to this, just something to consider.
• Line 60: Change to “2017) and areas with greater wave energy are likely to need more highly engineered structures.”
• Line 65: Change to “Nevertheless, the absence of an assessment..”
Methods:
• Line 80-85: More detail is needed here on the individual sites. I would like to see a table that details the age of the living shorelines, structural materials used, width of marsh, average tidal amplitude, etc. This should be easily compiled from the supplementary data.
• Line 86-87: A recent citation that might be helpful here showing that the vast majority of living shoreline studies have investigated fewer than 5 living shoreline sites: Smith, C., Rudd, M., Gittman, R.K., Melvin, E., Patterson, V., Renzi, J., Wellman, E. and Silliman, B., 2020. Coming to terms with living shorelines: A scoping review of novel restoration strategies for shoreline protection. Frontiers in Marine Science, 7, p.434.
• Lines 95-96: How many cores were taken per transect? I think its only 1 but I would be explicit here. This is also a little confusing- saying that you collected cores along transects would lead me to believe that you collected multiple cores per transect. Were the cores a consistent distance from the marsh edge?
• Lines 118-119: If the traps were set at high tide and puled at low tide wouldn’t they have a longer average soak time (i.e. ~6hrs)?
• Lines 122-123: Again, this seems like a short soak time, though twice as long as the minnow traps despite being deployed for the same tidal range
• Lines 127-129: What about fish? Did you only use the fyke nets to look at crabs?
• Lines 129-131: I don’t really understand this methodology, so this needs to be clearer. Were you estimating the area of the entire marsh or just what was being sampled by the nets?
• Line 137: Change to “Blue crabs”
• Line 139: What does counted and weighed in composite by sampling effort mean? Why aren’t all of the nekton measures standardized to effort?
• Lines 168-170: How long were the cameras deployed for during each survey event at each site?
• Lines 174-176: 3 to 5 cameras where- at each site? Across all sites? During each sampling event?
• Lines 168-181: It would be useful to give an overall range of the total number of days that were sampled at each site
• Line 212: Can you add a sentence here describing why you chose the z-score approach. I’m guessing its because there was really high variability among the difference pairs of sites?
Results:
• Line 241: Change to: “Among the four metrics (percent organic matter, C, N, and P), all paired…”
• Line 252: I think it should be |z| in parentheses
• Lines 283-285: Can you/did you look at the individual functional equivalence scores over time. It is very possible that some metrics (i.e. soil composition) will change significantly over time, whereas others that develop more quickly may not, which could confound the overall trend. I think investigating this for all of the metrics could be a useful addition.
• Lines 283-285: It may not be significant, but looking at Figure 2 suggests to me that there is a trend of z-score increase with increasing age.
Discussion:
• Lines 287-288: I’m not sure that you can draw this conclusion based on your data. You only have 2(?) sites that are two years or younger at the time of sampling and the majority of your sites are significantly older than that. Since you are calculating metrics across all of your sites, your z-score estimates are influenced by the majority of older sites that might have higher z-scores. Figure 2 suggests that one of your young sites reached functional equivalence and the other had not.
• Lines 309-328: Interesting!
• Lines 365-368: nice!
• Lines 370: Change to: “lost from the degradation of natural fringing marshes”
Figures and Tables:
• Supplementary dataset: Meta-data doesn’t include the units for many of the measurements and I think the table is missing data for the mussels in only the marsh for living shorelines.

Reviewer 2 ·

Basic reporting

please see below

Experimental design

please see below

Validity of the findings

please see below

Annotated reviews are not available for download in order to protect the identity of reviewers who chose to remain anonymous.

---

## Round 0.2 · accepted · Accept

Thanks for the detailed and comprehensive review. Following the reviewers, I am happy to recommend your interesting work for publication.

Reviewer 2 ·

Basic reporting

Clear, short, to the point, well-referenced

Experimental design

well designed

Validity of the findings

conclusions well stated and follows the results